# Sound of Story: Multi-modal Storytelling with Audio

**Jaeyeon Bae**[1*]   **Seokhoon Jeong**[2*]   **Seokun Kang**[1]   **Namgi Han**[3]
**Jae-Yon Lee**[3]   **Hyounghun Kim**[12]   **Taehwan Kim**[12]

[1]Artificial Intelligence Graduate School
[2]Department of Computer Science and Engineering
[3]School of Liberal Arts
Ulsan National Institute of Science and Technology, Korea
{qowodussla, shjd0246, oraclemiso,
hng88, jlee2791, h.kim, taehwankim}   @unist.ac.kr

## Abstract

Storytelling is multi-modal in the real world. When one tells a story, one may use all of the visualizations and sounds along with the story itself. However, prior studies on storytelling datasets and tasks have paid little attention to sound even though sound also conveys meaningful semantics of the story. Therefore, we propose to extend story understanding and telling areas by establishing a new component called *background sound* which is story context-based audio without any linguistic information. For this purpose, we introduce a new dataset, called *Sound of Story (SoS)*, which has paired image and text sequences with corresponding sound or background music for a story. To the best of our knowledge, this is the largest well-curated dataset for storytelling with sound. Our SoS dataset consists of 27,354 stories with 19.6 images per story and 984 hours of speech-decoupled audio such as background music and other sounds. As benchmark tasks for storytelling with sound and the dataset, we propose retrieval tasks between modalities, and audio generation tasks from image-text sequences, introducing strong baselines for them. We believe the proposed dataset and tasks may shed light on the multi-modal understanding of storytelling in terms of sound. Downloading the dataset and baseline codes for each task will be released in the link: https://github.com/Sosdatasets/SoS_Dataset.

## 1 Introduction

Story understanding and telling is one of the important but challenging problems in natural language processing and machine learning, and there have been extensive studies on this problem for decades (Harrison et al., 2017; Kočiský et al., 2018). Many prior researches have focused on building models that understand stories such as utilizing pre-trained model (Chen et al.,

2022; Wang et al., 2023; He et al., 2023), event-centric understanding (Chen et al., 2021) and using multiway-sampler (Xu et al., 2023). Also, lots of approaches are proposed that aim to understand the stories within the provided text (Mostafazadeh et al., 2016; Fan et al., 2018; Akoury et al., 2020), image (Huang et al., 2016; Bensaid et al., 2021; Krojer et al., 2022) and video (Rohrbach et al., 2015; Xu et al., 2016; Tapaswi et al., 2016; Li et al., 2019; Bain et al., 2020; Huang et al., 2020).

Despite there being many benchmarks for story understanding and telling, most of them less focus on audio even if it is one of the key elements for humans to understand stories (Lang et al., 1998; Lake et al., 2017). Therefore, we propose to extend story understanding and telling areas by establishing a new component called *background sound* which is story context-based audio without any linguistic information. For this purpose, we introduce a novel dataset called the *Sound of Story (SoS)*, which aims to leverage audio information for story understanding, distinguishing it from other existing story or audio datasets.

To create the SoS dataset, all the modalities are extracted from Condensed Movie Dataset (CMD) (Bain et al., 2020) and Large Scale Movie Description Challenge datasets (LSMDC) (Rohrbach et al., 2015). Image and text data extracted from the same movie clip have their contexts and both of them have the possibility to infer the story from the same movie clip. And in the case of audio such as background sound and music, it has the ability to enrich storytelling by representing and recognizing the context of the situations (Eronen et al., 2005; Stowell et al., 2015). Due to these characteristics, audio, image, and text extracted from the same movie clip become correlated and complementary. So, our SoS dataset allows three modalities generated from the same movie clip to be used for story understanding as a pair. In addition, we propose new story under-

---

* These authors contributed equally to this work.

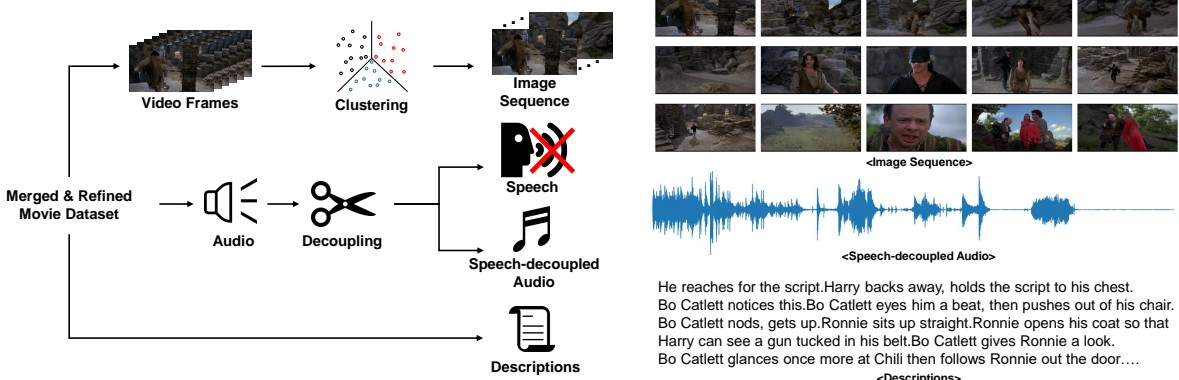

Figure 1: Overall data generating process flow for our Sound of Story (SoS) dataset. All the modalities are extracted from the same movie clips. We exclude the speech from audio to make it contain only pure audio data, which does not contain linguistic information.

standing benchmarks and their baseline models by introducing audio retrieval tasks and audio generation tasks using the SoS dataset.

These background sound tasks can be utilized in various applications. For example, for movies and cartoons, background sound is often produced according to a designated scene using various props or objects for a sense of reality. In the case of the retrieval task, among the sound sources made, it is possible to recommend the sound source which fits the best to the scene. Furthermore, background sound generation can produce more realistic audio while reducing human resource costs by directly generating the right audio for a given scene. The overall data generating process can be shown in Figure 1.

In summary, our contributions are as follows:

- We propose to extend story understanding and telling areas by establishing a new component called *background sound* which is story context-based audio without any linguistic information, which has not been well explored. For this purpose, we introduce a new dataset named *Sound of Story (SoS)* consisting of text and image sequences and audio for a story. To the best of our knowledge, this is the largest well-curated dataset for storytelling with sound.

- As benchmark tasks to show the significance and relevance of sound in a story, we introduce retrieval tasks between audio and other modalities and background sound generation tasks, and baseline models for each of them.

- We will release the dataset and code to generate them so that the community can generate more data for a variety of tasks to utilize sound in storytelling and story understanding research.

## 2 Related Work

**Story understanding and telling** Storytelling tasks pose significant challenges in the domain of natural language processing (NLP), and researchers have proposed various datasets and benchmarks to tackle these problems. Wikiplots, WritingPrompts (Fan et al., 2018), and ROCStories (Mostafazadeh et al., 2016) introduce datasets that revolve around text-based story plots. Wikiplots comprises a collection of approximately 113k story plots extracted from English Wikipedia. WritingPrompts (Fan et al., 2018) assembled their dataset by gathering pairs of human-written stories and corresponding prompts from online sources. ROCStories (Mostafazadeh et al., 2016) employed the use of Amazon Mechanical Turk (AMT) to generate short stories that incorporate common-sense knowledge, with a focus on everyday topics. These datasets are usually used for story understanding or story generation tasks.

In addition to text-based story plots, there exist numerous story datasets that incorporate images or videos. Visual Storytelling (Huang et al., 2016) introduces the first sequential vision-to-language dataset and explores its applicability in various tasks. It goes beyond simple object understanding or captioning for concrete scenes by considering input that encompasses temporal events. Similarly, ImageCoDe dataset (Krojer et al., 2022) also contains stories but is designed to enable models to

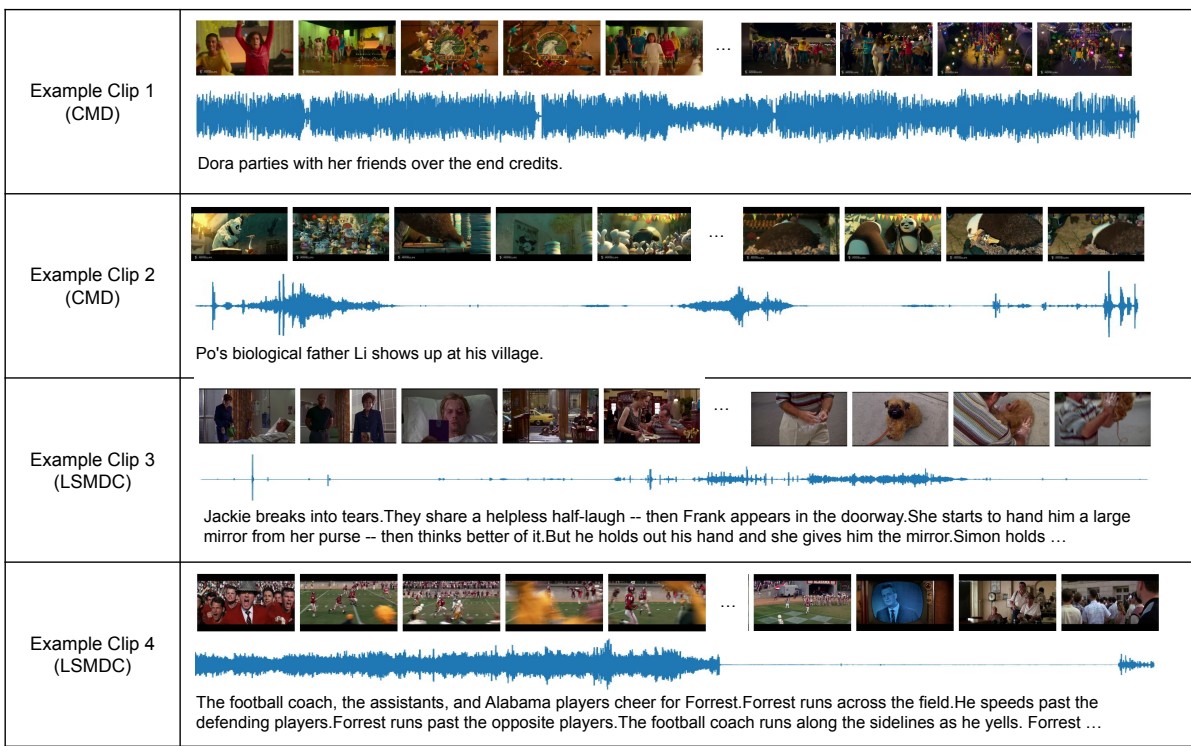

| Example Clip 1 (CMD) | | Dora parties with her friends over the end credits. |
| Example Clip 2 (CMD) | | Po's biological father Li shows up at his village. |
| Example Clip 3 (LSMDC) | | Jackie breaks into tears.They share a helpless half-laugh -- then Frank appears in the doorway.She starts to hand him a large mirror from her purse -- then thinks better of it.But he holds out his hand and she gives him the mirror.Simon holds … |
| Example Clip 4 (LSMDC) | | The football coach, the assistants, and Alabama players cheer for Forrest.Forrest runs across the field.He speeds past the defending players.Forrest runs past the opposite players.The football coach runs along the sidelines as he yells. Forrest … |

Figure 2: Example of SoS dataset. CMD and LSMDC have different lengths of descriptions.

learn detailed situations from finer visual differences within similar sets of images.

Movie data inherently encompasses storytelling. Various datasets such as MovieQA (Tapaswi et al., 2016) and MovieNet (Huang et al., 2020) have been collected by using movie data and have been used in various multimodal domains. Additionally, datasets like CMD (Bain et al., 2020) and LSMDC (Liu et al., 2019) not only provide movie videos but also include descriptions for the corresponding clips. This offers researchers an opportunity to explore models from a multimodal perspective. However, there are few datasets for story with sound. Existing benchmarks for audio mostly consist of music (Bertin-Mahieux et al., 2011; Sturm, 2012; Ferraro et al., 2021) or animal/environmental sounds (Salamon et al., 2014; Piczak, 2015; Gemmeke et al., 2017; Kim et al., 2019), and are designed to evaluate simple matching performance between objects (or sentiment) and audio input.

**Multi-modal retrieval tasks** Multi-modal retrieval models are trained to find similar features across different modalities by fusing. These models are tasked with understanding both the context of a given input and the interplay between features provided by other modalities. As a result, retrieval becomes a highly challenging and crucial task in order to comprehend the distinctive characteristics of multiple modalities.

The multi-modal retrieval task has been studied using various modalities such as image-text retrieval (Wang et al., 2020; Zhang et al., 2020; Cheng et al., 2022; Luo et al., 2022; Xuan and Chen, 2023), video-text (Ma et al., 2022; Gorti et al., 2022; Zhu et al., 2023), audio-image (Xu, 2020; Yang et al., 2022; Nakatsuka et al., 2023), video-audio (Surís et al., 2018; Gu et al., 2023; Cheng et al., 2023) and audio-text (Kim et al., 2022; Xin et al., 2023). Particularly, CLIP4CLIP (Luo et al., 2022), which performs well in the video-text retrieval task by calculating the similarities between the features of each modality obtained from the encoder, and X-CLIP (Ma et al., 2022) expands CLIP4CLIP and proposes a multi-grained regulation function to improve performance. Furthermore, Wav2CLIP (Wu et al., 2022) conducts retrieval for audio-image modalities based on CLIP architecture (Radford et al., 2021), while CLAP (Elizalde et al., 2023) focuses on audio-text retrieval. In this paper, we propose a story-based audio retrieval task for retrieving proper modal features like image sequences or descriptions.

**Multi-modal audio generation** There have been prior studies on generating audio by receiving cer-

| Dataset | Audio Type | # Story | # Images / Story | # Audio Hours | Avg. text length | Text Type |
|---|---|---|---|---|---|---|
| ImageCode | - | 9,402 | 17 | - | 23.3 | Captions |
| VIST | - | 50,136 | 4.2 | - | 10.2 | Description |
| Video Storytelling | open | 105 | - | 22 | 162.6 | Captions |
| MovieNet | open | 1,100 | - | 633 | 2004 | Description |
| MSR-VTT | open | 10,000 | - | 41.2 | 9.6 | Description |
| Ours | speech-decoupled audio | 27,354 | 19.6 | 984 | 88 | Description |

Table 1: Statistics compared to different storytelling datasets. In the audio types, "open" represents raw audio from the video, and "speech-decoupled audio" represents the speech-excluded audio. In the text types, "Caption" represents the text that explains only the image, and "Description" represents the text that includes the story.

tain modality information as input (Vasquez and Lewis, 2019; Yu et al., 2022; Borsos et al., 2022; Neves et al., 2022; Schneider et al., 2023; Yang et al., 2023; Agostinelli et al., 2023). Among several audio generation tasks, recently text-audio generation has been actively studied. AUDIO-GEN (Kreuk et al., 2022) uses an augmentation technique that mixes different audio samples when generating audio samples from captions. Riffusion (Forsgren and Martiros, 2022) fine-tunes the model to generate Spectrogram images using the image-to-text based on Stable Diffusion (Rombach et al., 2022). MUSICGEN (Copet et al., 2023) uses a single-stage language model with efficient token interleaving patterns and an unsupervised melody conditioning method. In this paper, we are interested in generating relevant audio given a story.

## 3 Dataset

We propose a novel dataset named the SoS dataset with 27,354 audio-image-text pairs. 22,330 pairs are created from CMD (Bain et al., 2020) and 5,024 from LSMDC dataset (Rohrbach et al., 2015). We extract the image and audio from the same video and pair them with the description accordingly so that the images and text share the same storyline. In the case of audio, it is related to the other modalities due to being extracted from the same clip so can be used in complementary. The examples of the SoS dataset are shown in Figure 2.

### 3.1 Movie Clips

CMD (Bain et al., 2020) provides the key clips and their high-level semantic descriptions from 3,605 movies. Each key clip has a duration of around 150 seconds and can be automatically obtained from YouTube, making it easily accessible. LSMDC (Rohrbach et al., 2015) consists of 118,081 short clips extracted from 202 movies. Each clip in LSMDC has a duration of around 3

to 12 seconds and is accompanied by captions generated from Description Video Services (DVS). In our work, we merge these two movie clip datasets to create the audio-image-text paired dataset. To bridge the gap between CMD and LSMDC, we adapt the LSMDC dataset to match the format of CMD. Since the short movie clips in LSMDC are contiguous, we concatenate them in sequential order until their combined length is similar to that of CMD. Specifically, we concatenate around 20 short movie clips together to create the raw movie clip data.

### 3.2 Audio processing

To facilitate audio processing, we first utilized ffmpeg (Tomar, 2006) to extract audio from the video, ensuring that the length of the extracted audio matched that of the raw movie clip video. Subsequently, we employed the *bytesep* framework proposed by (Kong et al., 2021) to decouple speech from mixed audio containing various sounds. The results of decoupled sounds may be considered as audio data that share the storyline with the raw movie clip video without any linguistic information. Speech-decoupling is proceeded because the speech itself includes the storyline and the model can learn the storyline focused on the speech, not the audio which is not our expectation. Duration distribution of the extracted audios can be seen in Figure 3.

### 3.3 Image processing

Since videos include many duplicated and similar images, only the key salient images that may represent storyline were extracted through image clustering. We conduct image clustering [1] using image fingerprints, which are representations for uniquely identifying an image generated based on the visual features. From all image fingerprints

---

[1] https://github.com/elcorto/imagecluster

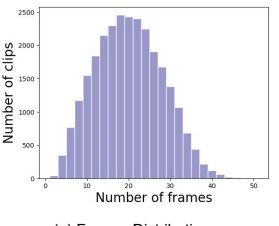 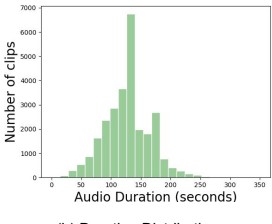 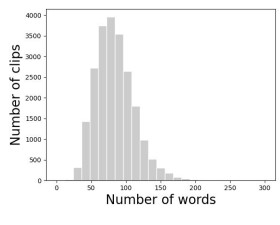 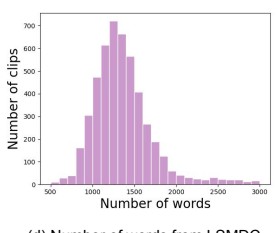

(a) Frames Distribution  (b) Duration Distribution  (c) Number of words from CMD  (d) Number of words from LSMDC

Figure 3: Distributions of each modality. (a) the number of extracted images from a movie video; (b) the duration of decoupled and processed audio for each story (c) the number of words in each story from CMD (d) the number of words in each story from LSMDC. Note that due to video clip concatenation to match video lengths, descriptions have larger lengths for stories from LSMDC. This text variation leads to the model requiring high-level text understanding.

extracted from frames of the movie clips, we gradually removed images with a fingerprint similarity of greater than 0.5 which was set empirically found. Through this process, fewer frames for static and more frames for dynamic scenes could be retained. The extracted image sequences can represent the entire video by capturing salient scenes with significant variations, even though it does not include all of the frames. By utilizing this approach, we can create essential image sequences of the video by focusing on scenes with meaningful changes. These processed image sequences provide a novel dataset that highlights the storyline with fewer frames and differentiates it from traditional video datasets. Figure 3 shows the distribution of extracted frames for each movie clip.

## 3.4 Text processing

There exist slight differences in clip lengths between CMD and LSMDC even though both of them provide short descriptions for each movie clip. Because we concatenate LSMDC movie clips to match the length of CMD movie clips, descriptions also must be formatted by concatenating the same number of descriptions with the video. As a result, the concatenated descriptions exhibit variations in sentence lengths as in Figure 3.

Furthermore, there are also inherent differences between the two datasets in how the descriptions were generated. CMD utilizes descriptions generated by users from YouTube, while LSMDC employs descriptive video services to generate descriptions. Because of this text variation difference between the two datasets, it makes our task more challenging by requiring high-level text understanding. We can check that the descriptions are well-aligned with other modalities in Figure 2.

## 3.5 Data Statistics and Analysis

Our SoS dataset is derived from 3,807 movies covered in the CMD and LSMDC datasets, resulting in a total of 27,354 clips. After processing, each clip has an average of 20 images, and the total number of images in our dataset is 535,707. The accumulated length of the extracted audio is around 984 hours, and each audio consists of around 150 seconds on average. Each clip's text description consists of 88 words on average. If divided into two different datasets, CMD and LSMDC consist of an average of 83.2 words and 1,378.7 words, respectively, with medians of 80 and 130.8, respectively. Quantitative comparison with other story-based datasets is shown in Table 1, and it can be seen that our proposed SoS dataset is the largest well-curated storytelling dataset with extracted audios.

We generate paired decoupled audio data that can be used complementarily for story understanding and telling with the image and text in the SoS dataset. The extracted decoupled audio, which is in-the-wild audio, contains abundant information. Specifically, we decoupled speech information to leave only pure audio consisting of background sound on the audio dataset. These remaining audios contain a mixture of various daily-life sounds and background music. This distinctive characteristic sets it apart from other audio datasets, making it an adequate audio dataset for story understanding.

## 4 Experiment

### 4.1 Retrieval Tasks

In this section, we introduce four retrieval tasks: audio-to-video, audio-to-text, text-to-audio, and video-to-audio retrieval using our proposed dataset. The retrieval tasks focus on finding proper audio

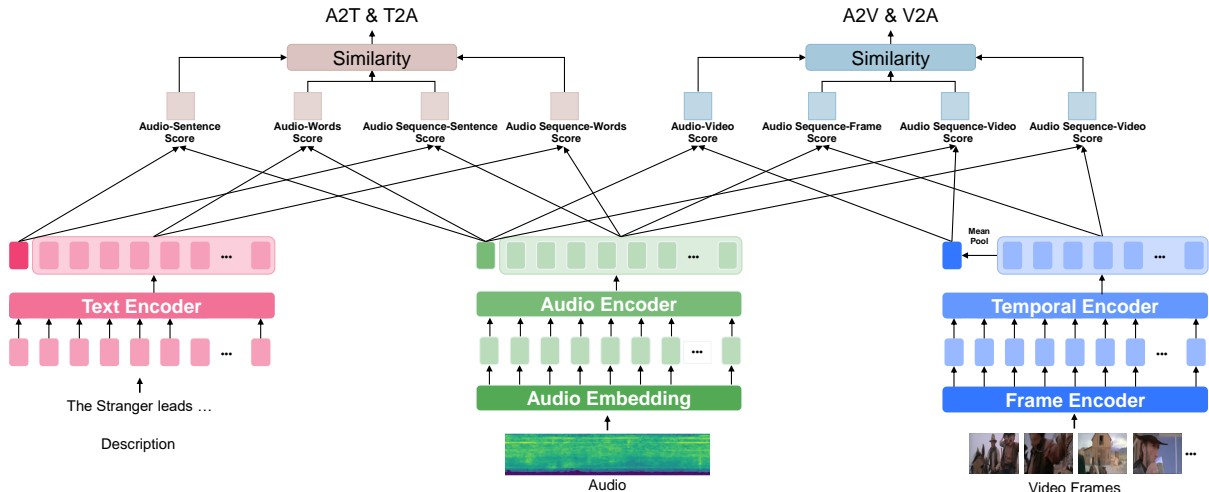

Figure 4: Proposed retrieval model architecture. Encoded features are obtained from the encoder corresponding to each modality. Features are divided into context-level features and sequence-level features. Then we calculate the score for each task and use the score to calculate the final similarity $Sim$.

corresponding to image sequences or text information containing a story, or matching image sequence or text given audio, which may be used as recommendation for background sound suitable for image sequence or storyline.

### 4.1.1 Retrieval Model

In this section, we introduce the model used in our proposed retrieval task. We are inspired when we create a baseline model in that the *Attention Over Simplicity Matrix* method proposed by X-CLIP (Ma et al., 2022) can be flexibly applied to Audio as well as Video-Text. Our overall architecture is illustrated in Figure 4.

For descriptions, we use the CLIP text encoder $E_T$ (Radford et al., 2021). The information obtained from the $E_t$ is defined as sentence feature $f_s$ and word feature $f_w$. In the case of video, the CLIP image encoder $E_I$ is used and the temporal encoder $E_t$ is added to obtain the temporal sequence feature. The obtained feature is defined as frame feature $f_f$, and the mean pooling of the frame feature is defined as video feature $f_v$. And in the case of the audio, audio information is extracted by performing audio embedding on Log Mel-filter bank coefficients and fed into the $E_A$ using pretrained Audio Spectrogram Transformer (Gong et al., 2021). In this case, the first token among the acquired feature is defined as audio $f_a$, and the remaining tokens are defined as audio sequence feature $f_{as}$.

In the case of audio-video retrieval, we first calculate the similarity between audio-video($S_{a,v}$), audio-frame($S_{a,f}$), audio sequence-video($S_{as,v}$),

and audio sequence-frame($S_{as,f}$). These similarities are calculated following Equation 1 to 4:

$$S_{a,v} = (f_a)^T(f_v) \in \mathbb{R}^1 \tag{1}$$

$$S_{a,f} = (f_f f_a)^T \in \mathbb{R}^{1 \times m} \tag{2}$$

$$S_{as,v} = f_{as} f_v \in \mathbb{R}^{n \times 1} \tag{3}$$

$$S_{as,f} = (f_{as})(f_f)^T \in \mathbb{R}^{n \times m} \tag{4}$$

where $m$ is the number of video frames and $n$ is the number of audio sequences. Except for audio-video similarity, an aggregation process is required to make it an instance-level similarity. Therefore, the audio-frame and audio sequence-video similarity is calculated through the following equations:

$$S'_{a,f} = \sum_{i=1}^{m} \frac{exp(S_{a,f(1,i)})}{\sum_{j=1}^{m} exp(S_{a,f(1,j)})} S_{a,f} \tag{5}$$

$$S'_{as,v} = \sum_{i=1}^{n} \frac{exp(S_{as,v(i,1)})}{\sum_{j=1}^{n} exp(S_{as,v(j,1)})} S_{as,v} \tag{6}$$

The audio-level similarity($S'_{aud}$) and video-level similarity($S'_{vid}$) are calculated to obtain the audio sequence-frame instance-level similarity:

$$S_{aud} = \sum_{i=1}^{n} \frac{exp(S_{as,f(i,*)})}{\sum_{j=1}^{n} exp(S_{as,f(j,*)})} S_{as,f(i,)} \in \mathbb{R}^{1 \times m} \tag{7}$$

$$S'_{aud} = \sum_{i=1}^{m} \frac{exp(S_{aud(1,i)})}{\sum_{j=1}^{m} exp(S_{aud(1,j)})} S_{aud(1,i)} \tag{8}$$

$$S_{vid} = \sum_{i=1}^{m} \frac{exp(S_{as,f(,i)})}{\sum_{j=1}^{m} exp(S_{as,f(,j)})} S_{as,f(,i)} \in \mathbb{R}^{n \times 1} \tag{9}$$

$$S'_{vid} = \sum_{i=1}^{n} \frac{exp(S_{vid(i,1)})}{\sum_{j=1}^{n} exp(S_{vid(j,1)})} S_{vid(1,1)} \tag{10}$$

| Eval | A2V | | V2A | | A2T | | T2A | | Random |
|---|---|---|---|---|---|---|---|---|---|
| | Ours | Wav2CLIP | Ours | Wav2CLIP | Ours | CLAP | Ours | CLAP | |
| R@1(↑) | **7.615** | 4.231 | **6.462** | 5.154 | **7.462** | 6.538 | **7.077** | 5.769 | 0.076 |
| R@5(↑) | **18.682** | 12.615 | **17.923** | 16.077 | **17.462** | 17.231 | **19.385** | 16.077 | 0.378 |
| R@10(↑) | **28.000** | 19.538 | **27.000** | 23.462 | **24.692** | 24.615 | **25.692** | 23.538 | 0.750 |
| Median R(↓) | 40.615 | **31.231** | 38.923 | **33.692** | **33.000** | 34.615 | **32.769** | 34.385 | 631.800 |
| Mean R(↓) | **32.000** | 52.000 | **33.000** | 47.000 | 60.000 | **52.000** | 63.500 | **53.500** | 639.132 |

Table 2: Retrieval performances with the proposed SoS dataset, which are measured on 1300 test samples. Random means the average of the results of five random retrievals.

where * denotes all elements in that dimension.

In addition, $S'_{as,f}$ are obtained from the means of $S'_{aud}$ and $S'_{vid}$ obtained by Equation 8 and 10. Therefore, the similarity between video and audio is obtained as the average of the obtained similarities, as shown in Equation 11.

$$Sim_{AV} = (S_{a,f} + S'_{a,f} + S'_{as,v} + S'_{as,f})/4 \quad (11)$$

For the audio-text retrieval, the similarity $Sim_{AT}$ can be calculated by replacing video-related features with text-related features.

### 4.1.2 Implementation Details

Of the 27,354 stories in our SoS dataset, 26,054 are split into train and 1,300 into test sets. In the case of video, a total of 40 images are used as input. At this time, if there are fewer than 40 images, 40 images are filled with duplicate sampling in accordance with the time order, and if there are more than 40, 40 images are obtained by uniform sampling. In the case of audio, the entire audio is used for learning, and random cropping is performed at this time. In the test, the same audio can be used in all tests by cropping and using the information-rich area of the audio. In the case of text, after tokenizing the description with a CLIP tokenizer, we crop 77 tokens from the front and use them similar to other CLIP-based models (Luo et al., 2022; Ma et al., 2022; Li et al., 2023). The batch size is 128 in the audio-video and 256 in the audio-text retrieval task, respectively. We conducted the experiment using 2 NVIDIA A100 80GB GPUs using the PyTorch library for 100 epochs, which took about 20 hours.

### 4.1.3 Results and Analysis

To evaluate the performance of the proposed baseline model with our SoS dataset, we compare it with Wav2CLIP (Wu et al., 2022) for A2V and V2A, and CLAP (Elizalde et al., 2023) for A2T and T2A. Both Wav2CLIP and CLAP are fine-tuned to our SoS dataset from their pre-trained weights. For the evaluation metrics, we utilize Recall@1(R@1), Recall@5(R@5), Recall@10(R@10), Mean Rank(Mean R), and Median Rank(Median R). Table 2 shows the performances for audio-to-video (A2V), video-to-audio (V2A), audio-to-text (A2T), and text-to-audio (T2A).

For A2V and V2A, each story of the SoS dataset consists of fairly long image sequences by using clustering from the video which has an average length of 215 seconds, unlike other datasets. In addition, our audio dataset includes various sounds to express a story and thus can better represent storylines compared to audio composed of simple sounds.

However, it also presents greater challenges. Therefore, compared to retrieving tasks using short videos and simple audio, it can be considered a challenging task. Nevertheless, with R@1 of 7.615 and 6.462 for both A2V and V2A, respectively, our baseline model performed better than its comparative model, Wav2CLIP, and showed reasonable evaluation results.

In addition, our text data consists of a storyline that implicitly represents the whole sequence, not a description of the scene, so it is challenging because the model has to understand the context of the scenes rather than a simpler understanding of the scene. Therefore, looking at the R@1 performance 7.462 and 7.077 in A2T and T2A, respectively, the performance of our proposed baseline model is better than the comparison model CLAP. Also, the results of retrievals can be found in our supplementary material or GitHub link [2].

### 4.2 Audio Generation Task

Composing and arranging background music and sound effects that exquisitely match the current scene's story is a highly challenging task that even human creators struggle (Thom, 1999). We propose an audio generation model called *SoSGen*, which

---

[2]https://github.com/Sosdatasets/SoS_Dataset

is designed to generate appropriate audio for the movie scene, given the scene's descriptions, frames, or both as the condition.

### 4.2.1 Audio Generation Model

SoSgen takes advantage of the architecture of diffusion models (Sohl-Dickstein et al., 2015), which learn a data distribution through gradual denoising. Stable diffusion models (Rombach et al., 2022), a variant of diffusion models, enable effective model conditioning by training the model to learn latent features. Following the same approach, we train SoSgen as a text-to-audio spectrogram diffusion model, using the following objective function:

$$\mathcal{L} = \mathbb{E}_{x,\epsilon\sim\mathcal{N}(0,1),t,c} \left[ \|\epsilon - \epsilon_\theta(\tilde{z}_t, t, c(d))\|_2^2 \right] \quad (12)$$

where $c$ is a trainable CLIP text encoder and $d$ is the SoS description.

For image+text-to-audio training, we follow ControlNet (Zhang and Agrawala, 2023) architecture, a lightweight and powerful method that enables additional conditioning of a diffusion model by freezing the original model's parameters. We freeze the text-to-audio model's parameters and resume training using a similar objective function:

$$\mathcal{L} = \mathbb{E}_{x,\epsilon\sim\mathcal{N}(0,1),t,c,c_{img}} \\ \left[ \|\epsilon - \epsilon_\theta(\tilde{z}_t, t, c(d), c_{img}(f))\|_2^2 \right] \quad (13)$$

where $c_{img}$ is an additional CLIP image encoder that embeds the SoS frame sequence $f$.

### 4.2.2 Implementation Details

Audio in movies rarely remains consistent until the end of a single scene. Also, using long audio as an input or output for a deep learning model is burdening in terms of hardware capacity. Thus, we slice the audio data into a length of approximately 15 seconds. Since extracted frames partially reflect semantic changes within the scene, audio slicing was performed treating audio data between two extracted frames as a base unit. The unit audios are accumulated and concatenated until their cumulative duration exceeds 15 seconds, ensuring that each sliced audio is never longer than 15 seconds. Audios that are originally longer than 15 seconds are not sliced because it implies the absence of significant visual changes for the duration, suggesting a high probability of long, consistent audio. Sliced audio files containing only a little sound were eliminated using the PyDub[3] library.

---

[3] https://github.com/jiaaro/pydub

To this end, we assessed the mean frequency of each segment and excluded any with a frequency below 78 Hz, a threshold determined through manual inspection.

This process conducted 185,341 audio data of roughly 15 seconds each, which were then converted into spectrograms. From this, we randomly split the dataset into 184,462 and 879 as training and testing, respectively.

For our base model, we leveraged Riffusion (Forsgren and Martiros, 2022), a checkpoint of Stable Diffusion v1-5 (Rombach et al., 2022), fine-tuned on pairs of music descriptions and spectrograms. Following the same approach, we fine-tuned the diffusion model from the Riffusion checkpoint with pairs of SoS descriptions and corresponding sliced audio spectrograms. In addition, we further trained the model in the setting of image and text conditioning, utilizing the ControlNet (Zhang and Agrawala, 2023) architecture. We use a sequence of frames extracted through clustering that correspond to the sliced audio as the image condition. SoS$_{text+image}$ was additionally trained from SoS$_{text}$ checkpoint, while SoS$_{image}$ resumed from the Riffusion checkpoint and used a redundant text condition in both training and testing. The specific model architecture can be found in Figure 5.

For training our models, we used a batch size of 1, 16 gradient accumulation steps, an initial learning rate of 1e-6, and a cosine learning rate scheduler. We fine-tuned the text-to-audio model for 406,000 steps and additional 525,000 steps to further train the text+image-to-audio model. Separately, the image-to-audio model was trained for 525,000 steps. We ran fine-tuning for 2 days and ControlNet training for another 2 days on a single NVIDIA GeForce RTX 3090 GPU.

### 4.2.3 Results and Analysis

**Baselines** We compare SoSgen to two baselines: Riffusion (Forsgren and Martiros, 2022) and MU-SICGEN (Copet et al., 2023), the state-of-the-art open source text-to-music generation model. The performance of the models is evaluated based on the Fréchet Audio Distance (FAD) (Kilgour et al., 2019). FAD is a reference-free evaluation metric that is computed using means and covariances of VGGish (Hershey et al., 2017) embeddings. We used a lightweight pytorch implementation[4] of FAD for more efficient evaluation.

---

[4] https://github.com/Sosdatasets/SoS_Dataset

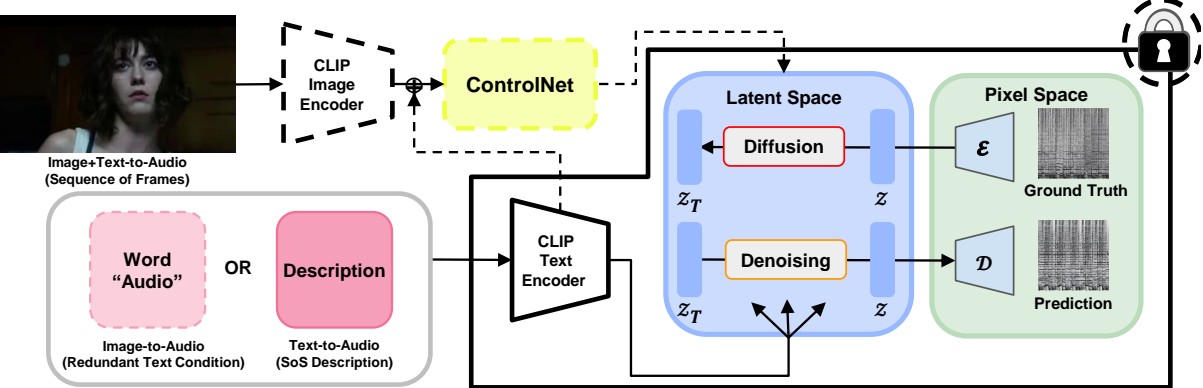

Figure 5: SoSgen architecture for training. The components and arrows drawn with dash lines are only used in the ControlNet-involved training, i.e. image-to-audio and image+text-to-audio. To train a text-to-audio model, we condition the denoising model with CLIP text embeddings of SoS descriptions. Once the text-to-audio model is sufficiently trained, we freeze the model and add image embeddings as a condition to train the text+image-to-audio model using ControlNet. Alternatively, an image-to-audio model can be trained from the Riffusion checkpoint, using a redundant text condition "Audio" for each training data. $\mathcal{E}$ denotes the encoder, $\mathcal{D}$ denotes the decoder, $\mathcal{Z_T}$ denotes latent representation at timestep $\mathcal{T}$.

Table 3: FAD (lower is better) of SoSgen and the baselines. SoSgen$_{text+image}$ is the only model given with the image and text condition. The rest are only given with SoS descriptions as the condition.

| Model | $FAD_{VGGish} \downarrow$ |
|---|---|
| Riffusion | 20.114 |
| MUSICGEN | 11.680 |
| SoSgen$_{text}$ | 10.935 |
| SoSgen$_{image}$ | 18.324 |
| SoSgen$_{text+image}$ | **9.099** |

Table 3 shows the comparison of SoSgen against Riffusion and MUSICGEN. SoSgen$_{text}$, Riffusion, and MUSICGEN generate an audio spectrogram given descriptions. For SoSgen$_{image}$ and SoSgen$_{text+image}$, an image sequence is also given as an additional condition.

The results indicate that SoSgen$_{text}$ and SoSgen$_{text+image}$ outperform Riffusion and MUSIC-GEN in terms of FAD. Although SoSgen$_{image}$ only shows a slight improvement from Riffusion, combining two modalities exceptionally enhances audio generation capability. The generated audio samples can be listened to in our GitHub link [5] and are also included in the supplementary material. The samples show SoSgen's superior performance on story audio generation tasks.

[5]https://github.com/Sosdatasets/SoS_Dataset

## 5 Conclusion

In this paper, we emphasize that sound also conveys meaningful semantics of the story. Thus, we propose a novel dataset named Sound of Story (SoS) dataset, audio-image-text pair dataset. One main feature of our dataset is that all the linguistic information inside the audio is eliminated. We named the audio without any linguistic information as "Background sound".

Background sounds can be utilized in various applications and we propose two related tasks, retrieval task and background sound generation task. To give an example of using it from videos such as movies and cartoons, background sound is often produced according to a designated scene using various props or objects for a sense of reality. In the case of the retrieval task, among the different sound sources made, it is possible to recommend the sound source which fits the best to the scene. Furthermore, background sound generation can produce more realistic audio while reducing human resource costs by directly generating the right audio for a given scene. To the best of our knowledge, it is the first attempt to do multimodal tasks with the speech-decoupled background sound relevant to the storyline.

## Limitations

To use audio data in story understanding, there are some limitations to be solved. First, we create audio data by decoupling the speech to exclude lin-

guistic information and preserving only pure audio. However, the generated audio may be yet distant from what people encounter in daily life because there is a possibility that various noises caused by the generation from the movie clip remain. Second, the audio-image-text pair is strongly connected, but it is not perfect due to the nature of the audio. Since audio includes a lot of subjective elements, various correct answers may exist for one story. To deal with these limitations, more strict and cautious policies to process not only audio but also image and text for story understanding should continue to be studied as future work.

## Ethics Statement

One potential ethical concern from our proposed dataset is the ability of our model to generate audio resembling existing work as it may potentially pose copyright issues such as regarding background music. In addition, generated audio may be too uncomfortable or inappropriate to hear.

## Acknowledgements

This work was partly supported by Institute of Information & communications Technology Planning & Evaluation (IITP) grant funded by the Korea government (MSIT) (No.2022-0-00608, Artificial intelligence research about multi-modal interactions for empathetic conversations with humans & No.2021-0-02068, Artificial Intelligence Innovation Hub) and the National Research Foundation of Korea(NRF) grant funded by the Korea government(MSIT) (No. RS-2023-00219959).

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
