# OpenReview forum: "Sound of Story: Multi-modal Storytelling with Audio"
_EMNLP/2023/Conference — EMNLP 2023 Findings_

### Official Review · Reviewer_s1GB · 2023-08-04

**Soundness:** 4

**Excitement:**

4: Strong: This paper deepens the understanding of some phenomenon or lowers the barriers to an existing research direction.

**Paper Topic And Main Contributions:**

The authors curate a large multi-modal story dataset, Sound of Story (SoS), with the focus on the audio part of the story. The dataset of 27,354 stories is built based on two movie-videos datasets, CMD and from LSMDC datasets. The audio parts of SoS are decoupled with the speech to prevent leaking language information for the retrieval task benchmark of this paper.

In the second part of the paper, the author proposed two tasks: cross-modality (audio <-> video and audio <-> text) retrieval tasks and the audio generation task. The baseline results on retrieval tasks show reasonable performance on Recall@1 with at least 5% compared to random baseline of 0.07%.

As for audio generation tasks, the demo website also presents examples of good generated audio clips with multiple baseline models, e.g., SoSgen.

The contributions of this paper are two-fold: (1) curation of the first large audio-focused multi-modal story dataset, and (2) proposing benchmarks of cross-modal retrieval tasks and the audio generation task.


**Questions For The Authors:**

It’s not clear how important is the speech-decoupling process? What portion of the movie clips include speech signals?

What’s the rationale of “we crop 77 tokens from the front and use it.” in line 370


**Reasons To Accept:**

The paper has an unique focus on speech-decoupled audio for the multi-modal story dataset.
The demo website shows a promising performance of the baseline models. It’s already an interesting experience to explore benchmark results .



**Reasons To Reject:**

The genre of audio clips are not clear. It would be helpful to have a high-level categorization of audio clips. As a follow up concern, having such categorization can provide more information of the capability of current baseline models for error analysis.


**Reproducibility:**

5: Could easily reproduce the results.

**Reviewer Confidence:**

3: Pretty sure, but there's a chance I missed something. Although I have a good feel for this area in general, I did not carefully check the paper's details, e.g., the math, experimental design, or novelty.

---

> ### Author Rebuttal · Authors · 2023-08-29
>
> Dear reviewer s1GB, thank you for your positive feedback and sharing your insights. We address your comments below and will incorporate all feedback into our final paper.
>
> > Q.1. The genre of audio clips are not clear. It would be helpful to have a high-level categorization of audio clips. As a follow up concern, having such categorization can provide more information of the capability of current baseline models for error analysis.
>
> Thank you for sharing your interesting insight for our dataset. We also agree that the collected audio data will be helpful in many ways if it becomes a high-level categorization. However, unlike text and images, background sounds are ambiguous in performing categorization, so performing high-level categorization itself is another challenging part for background sound audio. For example, when the background sound is heard in the order of a car moving, a car hitting somewhere, a car door closing, and a running footstep, it is relatively difficult to distinguish which category of the background sound is compared to the image and text. However, as you mentioned, if meta-information related to categories can be generated and utilized, it will be an opportunity for the dataset to become more enabled, so we will explore it.
>
> > Q.2. It’s not clear how important is the speech-decoupling process? What portion of the movie clips include speech signals?
>
> Thank you very much for pointing out the important content. In the prior storytelling datasets, they usually perform various tasks based on image sequences, videos, or summarized story information. In contrast, in our paper, we would like to expand the storytelling area by establishing a new component called “background sound” which is story context-based audio without any linguistic information. To the best of our knowledge, it is the first attempt to do multimodal tasks with the speech-decoupled background sound with storyline and these are challenging and not trivial compared to other audio tasks.
> In our SoS dataset, 27,354 movie clips have 984 hours in total, and the audio extracted from them has the same amount of time. And when we checked the time that the audio split with speech and background sound, there were about 408 hours for the speech, and about 480 hours for the background sound. __It means that the background sound is a significant part of the video and is not trivial to use only background sound in storytelling tasks.__
>
> > Q.3. What’s the rationale of “we crop 77 tokens from the front and use it.” in line 370
>
>
>
> The reason why we crop in front of 77 tokens is to follow the setting of the x-clip[3] paper that we use as baseline. When using the pre-trained model provided by the Open AI CLIP paper[1], the maximum context length of the tokenizer is 77. And in the x-clip, the max-token-length is set to 77 and used accordingly. At this time, if the number of tokens exceeds 77, the x-clip uses the front 77 tokens in the sentence to perform several tasks, all of which show reasonable performance. It is a common technique to use front tokens for clip encoder if the sentence is too long and not only x-clip but also other papers with clip also use. (Clip4clip [2], DeCap[4] etc). Based on these, our experiment also uses a setting that is used by crop in front of the sentence for sentences exceeding 77 tokens.
>
>
>
> __References:__
>
> > [1] Radford, et al. “Learning transferable visual models from natural language supervision” ICML 2021.
>
> > [2] Luo, et al.  “Clip4clip: An empirical study of clip for end to end video clip retrieval and captioning” Neurocomputing 2022
>
> > [3] Yiwei Ma, et al. “X-clip: End-to-end multi-grained contrastive learning for video-text retrieval” ACMMM22.
>
> > [4] Li, et al. “DeCap: Decoding CLIP Latents for Zero-Shot Captioning via Text-Only Training” ICLR 2023.

---

### Official Review · Reviewer_sX7X · 2023-08-05

**Soundness:** 3

**Excitement:**

3: Ambivalent: It has merits (e.g., it reports state-of-the-art results, the idea is nice), but there are key weaknesses (e.g., it describes incremental work), and it can significantly benefit from another round of revision. However, I won't object to accepting it if my co-reviewers champion it.

**Paper Topic And Main Contributions:**

This paper presents a new dataset for multimodal storytelling with audio. Especially, the paper proposes to remove the speech and only retain the speech decoupled audio so that the "linguistic information" in the audio is removed. In addition, the paper proposes two tasks on the dataset, one for multimodal retrieval and one for audio generation based on text and image.

**Reasons To Accept:**

1. The paper proposes a new dataset that might be good for the community. And the dataset and code will be made publicly available.
2. The related work section contains relevant information about multimodal storytelling , which could be beneficial to the community.

**Reasons To Reject:**

1. I failed to truly understand the motivation behind the dataset curation. Why is removing speech from the audio meaningful? To me it is a bit too ad hoc. I think, with the speech, the multimodal retrieval problem will be more meaningful because there are semantic multimodal alignment.
2. The proposed two tasks are also not very related to the SoS dataset, any video dataset could be used on the tasks.

**Reproducibility:**

4: Could mostly reproduce the results, but there may be some variation because of sample variance or minor variations in their interpretation of the protocol or method.

**Reviewer Confidence:**

3: Pretty sure, but there's a chance I missed something. Although I have a good feel for this area in general, I did not carefully check the paper's details, e.g., the math, experimental design, or novelty.

---

> ### Author Rebuttal · Authors · 2023-08-29
>
> Dear reviewer sX7X, thank you for your constructive feedback. We address your comments below and will incorporate all feedback into our final paper.
>
> > Q.1. I failed to truly understand the motivation behind the dataset curation. Why is removing speech from the audio meaningful? To me it is a bit too ad hoc. I think, with the speech, the multimodal retrieval problem will be more meaningful because there are semantic multimodal alignment.
>
> __The purpose of this paper is to expand the storytelling area by establishing a new component called “background sound” which is story context-based audio without any linguistic information.__ To the best of our knowledge, it is the __first attempt__ to do multimodal tasks with the speech-decoupled background sound with storyline and these are challenging and not trivial compared to other audio tasks. And because of the goal of our paper, speech audio is not for us to consider even if it could be helpful for the retrieval tasks. The meaning of removing speech from the audio will be answered in the response to Q2, which is closely related.
>
> > Q.2. The proposed two tasks are also not very related to the SoS dataset, any video dataset could be used on the tasks.
>
>
> To the best of our knowledge, it is the first attempt to use the story context-based background sound to the retrieval task. And __because the background sound is a new component in the tasks, it is different from other existing tasks.__ Also in the __audio generation task, the purpose is to create a story context-based background sound__ so it is not trivial. Existing audio generation models [1, 2, 3, 4] demonstrate their generation proficiency based on a direct description rather than a story-related context. For example, suitable inputs for the aforementioned models are natural language prompt “Jazz” or an image of a singing woman. In contrast, SoSgen is capable of generating appropriate audio for a story context, such as the prompt "Mrs. Henderson accidentally shuts off Fido's collar which leads to disastrous results" or the image sequence of the corresponding scene, which is an example that can be seen on our demo website. Such a task can only be accomplished using SoS dataset, since to our best knowledge, no previous dataset has provided pairs of story context and background sound.
>
> __These background sound tasks can be utilized in various applications.__ To give an example of how to use it from videos such as movies and cartoons, background sound is often produced according to a designated scene using various props or objects for a sense of reality. In the case of the retrieval task at this time, among the various sound sources made, it is possible to recommend the sound source which fits the best to the scene. Furthermore, background sound generation can produce more realistic audio while reducing human resource cost by directly generating the right audio for a given scene.
>
> __References__
> > [1] Felix Kreuk, Gabriel Synnaeve, Adam Polyak, Uriel Singer, Alexandre Défossez, Jade Copet, Devi Parikh, Yaniv Taigman, and Yossi Adi. 2022. Audiogen: Textually guided audio generation. arXiv preprint arXiv:2209.15352.
>
> > [2] Seth Forsgren and Hayk Martiros. 2022. Riffusion - Stable diffusion for real-time music generation.
>
> > [3] Jade Copet, Felix Kreuk, Itai Gat, Tal Remez, David Kant, Gabriel Synnaeve, Yossi Adi, and Alexandre Défossez. 2023. Simple and controllable music generation. arXiv preprint arXiv:2306.05284.
>
> > [4] Iashin, Vladimir, and Esa Rahtu. Taming visually guided sound generation. arXiv preprint arXiv:2110.08791 (2021).

---

### Official Review · Reviewer_Lmj8 · 2023-08-06

**Soundness:** 3

**Excitement:**

3: Ambivalent: It has merits (e.g., it reports state-of-the-art results, the idea is nice), but there are key weaknesses (e.g., it describes incremental work), and it can significantly benefit from another round of revision. However, I won't object to accepting it if my co-reviewers champion it.

**Missing References:**

[1] MAD: A Scalable Dataset for Language Grounding in Videos from Movie Audio Descriptions
: https://arxiv.org/abs/2112.00431

**Paper Topic And Main Contributions:**

The authors explore the problem of multimodal storytelling in movies with accompanying sound/background music by augmenting two standard datasets. Further they propose baseline models for two related tasks including multimodal retrieval and audio generation.

**Questions For The Authors:**

* Have the authors tried using a ViT-based encoder for extracting audio representations i.e. **AST [1]**, since it might be better suited for downstream retrieval tasks?
* In the case of the visual stream, have the authors tried extracting shots followed by clustering on shot-specific frames to determine the sequence of informative frames?

**[1]**  https://arxiv.org/abs/2104.01778

**Reasons To Accept:**

* Potential of the dataset to be used for visual scene-guided background score generation.
* Novel multimodal retrieval task for stories in media by considering all possible modalities including audio, visual, and text.

**Reasons To Reject:**

* In the retrieval part, the contributions are not clear in terms of the dataset construction. The authors primarily augment two datasets by extracting the audio separately from the videos and separating into speech and non-speech components.

* Details of score computation in Eq (1) and (2) are not mentioned. Further, for the sequence to single token similarity, it is not clear how the aggregation is performed after computing similarities with multiple token elements.

* The details in Figure 4 are confusing. It might be better to define the terms associated with the similarity scores (token to sequence similarity or sequence to sequence similarities)

* For the retrieval section, experimental results should be shown with pretrained multimodal audio-visual (**wav2CLIP[1]**) or audio-text encoders (**CLAP[2]**).

* In the audio generation setup, in section **4.2.2**, it is mentioned that two separate models are trained ($SOS_{image}$ and $SOS_{image+text}$). Whereas in section **4.2.1**, it is mentioned that the text model is frozen, and then the image guidance is performed on top of it ($SOS_{image+text}$ ?). It is not clear how $SOS_{image}$ is used in this setup?

**[1]** https://arxiv.org/pdf/2110.11499.pdf

**[2]** https://arxiv.org/abs/2206.04769

**Reproducibility:**

4: Could mostly reproduce the results, but there may be some variation because of sample variance or minor variations in their interpretation of the protocol or method.

**Reviewer Confidence:**

4: Quite sure. I tried to check the important points carefully. It's unlikely, though conceivable, that I missed something that should affect my ratings.

---

> ### Author Rebuttal · Authors · 2023-08-29
>
> Dear reviewer Lmj8, thank you for your thoughtful feedback. We address your comments below and will incorporate all feedback into our final paper.
>
> > Q.1. In the retrieval part, the contributions are not clear in terms of the dataset construction. The authors primarily augment two datasets by extracting the audio separately from the videos and separating into speech and non-speech components.
>
> In the prior storytelling datasets, they usually perform various tasks based on image sequences, videos, or summarized story information. In contrast, in our paper, __we would like to expand the storytelling area by establishing a new component called “background sound”__ which is story context-based audio without any linguistic information. To the best of our knowledge, it is the first attempt to do multimodal tasks with the speech-decoupled background sound with storyline.
>
> A typical example of this background sound retrieval task can be used is the production of videos such as movies and cartoons. When producing the above video, in many cases, to provide realistic sound, various objects or props are used to produce sound according to the scene and insert it according to the scene. This background sound retrieval task can help them select the best sound sequence when given a specific scene and a list of audio samples produced. This process may help them make their video more engaging.
>
> > Q.2. Details of score computation in Eq (1) and (2) are not mentioned. Further, for the sequence to single token similarity, it is not clear how the aggregation is performed after computing similarities with multiple token elements.
>
> We are afraid we could not give you enough explanation due to the page limit and thank you for asking us a good question to make up for the inconvenience. We adopted the Attention Over Similarity Matrix method proposed in X-CLIP [1] for Video-Text Retrievals to Audio-Video and Audio-Text Retrievals, as mentioned in the paper (lines 329 to 332).
>
> The equation (1) at line 356 used to perform Audio-Video Retrieval is obtained through the following formula. First, in the case of audio, audio feature $f_{a}$ and audio sequence feature $f_{as}$ are obtained through audio encoder $E_{A}$. For video, video feature $f_v$ and frame feature $f_{f}$ are obtained through Clip Image Encoder $E_{I}$ and Temporary Encoder $E_{t}$.
>
> Next, we calculate the similarity between each acquired feature. These are the equations for them:
>
> Audio-Video Similarity: $ S_{a,v}=(f_a)^T(f_v) \in \mathbb{R}^{1}$
>
> Audio-Frame Similarity: $S_{a,f}=(f_{f}f_{a})^{T} \in \mathbb{R}^{1 \times m}$
>
> Video-Audio Sequence Similarity: $S_{as,v}=f_{as}f_{v} \in \mathbb{R}^{n \times 1}$
>
> Audio Sequence-Frame Similarity: $S_{as,f}=(f_{as})(f_{f})^{T} \in \mathbb{R}^{n \times m}$
>
> where  $m$ is the number of video frames and $n$ is the number of audio sequences.
>
> Except for Audio-Video Similarity, an aggregation process is required to make it an instance-level similarity. Therefore, the Audio-Frame and Audio Sequence-Video Similarity is calculated through the following equation.
>
> $S'\_{a,f} = \sum\_{i=1}^{m} \frac{exp(S\_{a,f(1,i)})}{\sum\_{j=1}^{m}exp(S\_{a,f(1,j)})} S\_{a,f}$
>
> $S'\_{as,v} = \sum\_{i=1}^{n} \frac{exp(S\_{as,v(i,1)})}{\sum\_{j=1}^{n}exp(S\_{as,v(j,1)})} S\_{as,v}$
>
> Finally, the audio-level similarity and video-level similarity are calculated to obtain the Audio Sequence-Frame instance-level similarity. In notation, * means all elements in that dimension.
>
> audio-level similarity:
>
> $S\_{aud}=\sum\_{i=1}^{n} \frac{exp(S\_{as,f(i,*)})}{\sum\_{j=1}^{n}exp(S\_{as,f(j, *)})} S\_{as,f(i,\*)} \in \mathbb{R}^{1 \times m}$
>
> $S'\_{aud}=\sum\_{i=1}^{m}\frac{exp(S\_{aud(1,i)})}{\sum\_{j=1}^{m}exp(S\_{aud(1, j)})}S_{aud(1,i)}$
>
> video-level similarity:
>
> $S\_{vid}=\sum\_{i=1}^{m} \frac{exp(S\_{as,f(\*,i)})}{\sum\_{j=1}^{m}exp(S_{as,f(\*, j)})}S\_{as,f(\*,i)} \in \mathbb{R}^{n \times 1}$
>
> $S'\_{vid}=\sum\_{i=1}^{n}\frac{exp(S\_{vid(i,1)})}{\sum\_{j=1}^{n}exp(S\_{vid(j, 1)})}S\_{vid(1,1)}$
>
> In addition, $S'\_{as,f}$ are obtained from the means of $S'\_{aud}$ and $S'_{vid}$ obtained by the above equations.
> Therefore, finally, equation (1) of paper line 356 is obtained as the average of the similarities obtained above, as shown in the equation below.
>
> $Sim\_{AV}=(S\_{a,f}+S'\_{a,f}+S'\_{as,v}+S'\_{as,f})/4$
>
> In the case of Equation (2) used in Audio-Text Retrieve, it can be calculated from the above calculation formula by replacing video-related features with text-related features.
>
> We will include these details in supplementary material of our final paper.
>
>
> > Q.3. The details in Figure 4 are confusing. It might be better to define the terms associated with the similarity scores (token to sequence similarity or sequence to sequence similarities)
>
> We are sorry if our figure was confusing. What we wanted to provide in Figure 4 was to express the equation described in the response to Q.2. We will add the details in the revision.
>
> > Q.4. For the retrieval section, experimental results should be shown with pretrained multimodal audio-visual (wav2CLIP[1]) or audio-text encoders (CLAP[2]).
>
> Thank you for your suggestion. We conducted an additional retrieval experiment using the Wav2CLIP and CLAP you suggested. We finetune pretrained Wav2CLIP and CLAP on our dataset. First of all, in the case of our audio encoder, we did not use the pretrained model, but performed audio embedding with Log Mel-filter bank coefficients as mentioned in our paper (lines 342 to 345), and then used the pretrained ViT with conventional images. Therefore, unlike the audio encoders of Wav2CLIP and CLAP, our audio encoders have not been pretrained with audio. The results are shown in Table 1 below.
>
> __Audio-Video Retrieval__
> In the case of Audio-Video Retrieval, as shown in Table 1, ours showed mostly better performance than Wav2CLIP.
>
> __Audio-Text Retrieval__
> In the case of Audio-Text Retrieval, CLAP performs slightly better than ours, showing the benefit of using pretrained model of CLAP on audio data. On the other hand, we use AST model not pretrained on audio data but it still shows comparable performance compared to CLAP, which shows that our model may be a reasonable baseline as non-pretrained model. We will include the new results to our final paper. Thank you.
>
> |          |   A2V  | A2V |   |   V2A  | V2A |   |   A2T  |A2T|   |   T2A  |T2A|   |         |
> |:--------:|:------:|:--------:|---|:------:|:--------:|---|:------:|:------:|---|:------:|:------:|---|:-------:|
> | Eval     |  Ours  | Wav2CLIP |   |  Ours  | Wav2CLIP |   |  Ours  |  CLAP  |   |  Ours  |  CLAP  |   |  Random |
> | R@1      |  __7.615__ |   4.231  |   |  __6.462__ |   5.154  |   |  5.923 |  __6.538__ |   |  5.692 |  __5.769__ |   |  0.076  |
> | R@5      | __18.682__ |  12.615  |   | __17.923__ |  16.077  |   | 15.615 | __17.231__ |   | 14.692 | __16.077__ |   |  0.378  |
> | R@10     | __28.000__ |  19.538  |   | __27.000__ |  23.462  |   | 21.308 | __24.615__ |   | 22.308 | __23.538__ |   |  0.750  |
> | Median R | 40.615 |  __31.231__  |   | 38.923 |  __33.692__  |   | __30.231__ | 34.615 |   | __30.769__ | 34.385 |   | 631.800 |
> | Mean R   | __32.000__ |  52.000  |   | __33.000__ |  47.000  |   | 67.000 | __52.000__ |   | 69.000 | __53.500__ |   | 639.132 |
>
> Table 1. Additional retrieval results with Wav2CLIP and CLAP.
>
>
>
> > Q.5. In the audio generation setup, in section 4.2.2, it is mentioned that two separate models are trained (...)
>
> While we have referred to our model as "$SoSgen\_{image},$" its structure and training procedure are identical to "$SoSgen\_{text+image}$." As mentioned in the caption of Figure 5 in the main paper, the primary distinction between the two models lies in the text conditions given on training and inference time. Specifically, for "$SoSgen\_{image},$" we consistently employ a redundant textual condition, namely, the word "Audio." On the other hand, “$SoSgen\_{text+image}$” uses SoS descriptions as the text condition. This experimental setup was designed in order to validate the impact of each modality in the SoS dataset on the audio generation task.
>
> > Q.6. Have the authors tried using a ViT-based encoder for extracting audio representations i.e. AST [1], since it might be better suited for downstream retrieval tasks?
>
> We apologize for any confusion. Actually, __we use AST [2] to extract audio representations in our baseline model__. The explanation about the audio encoder mentioned in line 341 to 345 in the main paper is about how the AST works in our model but we missed to add the reference. Thank you and we will add the reference in revision.
>
> > Q.7. In the case of the visual stream, have the authors tried extracting shots followed by clustering on shot-specific frames to determine the sequence of informative frames?
>
>
> We assume that the question was about whether we have tried to manually divide the movie clip into shots, which are smaller units of scene, and do clustering to determine the sequence of informative frames.
>
> If our understanding is correct, no, we have not tried it. Actually, we extracted the frame sequence from the scene. As mentioned from line 238 to 245, we extract frames using image fingerprints similarity. And based on the technique and results mentioned from line 245 to 256, __we made this decision based on our own assessment that the frame sequence was adequately coherent and well-aligned with both the accompanying text and audio. The example of extracted frame sequences can be shown in the provided supplementary material.__ But, we agree that your suggestion is also relevant. Thank you for sharing your insight.
>
> __References:__
> > [1] Yiwei Ma, et al. “X-clip: End-to-end multi-grained contrastive learning for video-text retrieval” ACMMM22.
>
> > [2]Yuan Gong, et al.  “AST: Audio Spectrogram Transformer” Interspeech 2021

---

### Meta-Review · Area_Chair_2LX7 · 2023-09-15

**Recommendation:** 3

**Metareview:**

This paper introduces a new dataset of multimodal storytelling (images from the story, the text itself of the story, and sound effects/music that are part of the story. From the demo page, it seems as though the sound is predominantly music). The dataset is quite large at 27k stories with 19 images per story and 984 hours of sound, and is derived from Hollywood movies. The paper also presents some benchmark tasks for the dataset such as retrieval and audio generation from image/text.

Pros:
The dataset is novel in its purpose and scope, and is to be made publicly available

Cons:
-Some of the technical details for how similarity scores are not clear (see comments by reviewer Lmj8)
-The experiments don't include results from pre-trained models like CLAP or wav2Clip
-It is not clear how the proposed dataset offers any advantages over a dataset of videos + captions/narrations
-It is also not clear why speech/dialog was removed from the soundtracks. The author's response to reviewer sX7X on this point doesn't really explain what is more compelling about the music/background sound with the movie dialog removed, compared to the full original audio

---

### Decision · Program_Chairs · 2023-10-07

**Decision:**

Accept-Findings

**Comment:**

This paper introduces a new dataset of multimodal storytelling (images from the story, the text itself of the story, and sound effects/music that are part of the story. From the demo page, it seems as though the sound is predominantly music). The dataset is quite large at 27k stories with 19 images per story and 984 hours of sound, and is derived from Hollywood movies. The paper also presents some benchmark tasks for the dataset such as retrieval and audio generation from image/text.

Pros:
The dataset is novel in its purpose and scope, and is to be made publicly available

Cons:
-Some of the technical details for how similarity scores are not clear (see comments by reviewer Lmj8)
-The experiments don't include results from pre-trained models like CLAP or wav2Clip
-It is not clear how the proposed dataset offers any advantages over a dataset of videos + captions/narrations
-It is also not clear why speech/dialog was removed from the soundtracks. The author's response to reviewer sX7X on this point doesn't really explain what is more compelling about the music/background sound with the movie dialog removed, compared to the full original audio